# Hand Gesture Interface for Robot Path Definition in Collaborative Applications: Implementation and Comparative Study

**DOI:** 10.3390/s23094219

**Published:** 2023-04-23

**Authors:** Aleš Vysocký, Tomáš Poštulka, Jakub Chlebek, Tomáš Kot, Jan Maslowski, Stefan Grushko

**Affiliations:** Department of Robotics, Faculty of Mechanical Engineering, VSB—Technical University of Ostrava, 17. Listopadu 2172/15, 708 00 Ostrava, Czech Republic

**Keywords:** human-robot collaboration, human-robot interaction, hand tracking, hand recognition, gesture

## Abstract

The article explores the possibilities of using hand gestures as a control interface for robotic systems in a collaborative workspace. The development of hand gesture control interfaces has become increasingly important in everyday life as well as professional contexts such as manufacturing processes. We present a system designed to facilitate collaboration between humans and robots in manufacturing processes that require frequent revisions of the robot path and that allows direct definition of the waypoints, which differentiates our system from the existing ones. We introduce a novel and intuitive approach to human–robot cooperation through the use of simple gestures. As part of a robotic workspace, a proposed interface was developed and implemented utilising three RGB-D sensors for monitoring the operator’s hand movements within the workspace. The system employs distributed data processing through multiple Jetson Nano units, with each unit processing data from a single camera. MediaPipe solution is utilised to localise the hand landmarks in the RGB image, enabling gesture recognition. We compare the conventional methods of defining robot trajectories with their developed gesture-based system through an experiment with 20 volunteers. The experiment involved verification of the system under realistic conditions in a real workspace closely resembling the intended industrial application. Data collected during the experiment included both objective and subjective parameters. The results indicate that the gesture-based interface enables users to define a given path objectively faster than conventional methods. We critically analyse the features and limitations of the developed system and suggest directions for future research. Overall, the experimental results indicate the usefulness of the developed system as it can speed up the definition of the robot’s path.

## 1. Introduction

Collaborative robotics offers new possibilities for automating manufacturing processes. Collaboration between humans and robots brings benefits because it combines the advantages of both. Robots work fast, precisely, and consistently, while people are more flexible, creative, and better at decision-making [1]. There are several low-volume, high-value manufacturing processes that involve a large number of product variants and complex tasks, with great potential for automation using collaborative robotics [2]. These processes may require a frequent revision of the robot path. There is a potential for the development of HMIs (human–machine interfaces), which can speed up the process of adjustment of the robot path.

Standard robot programming requires skilled and trained operators. Teach pendants and touch screens are usually used for programming simple tasks or for debugging more complicated programmes. Operating the teach pendant and other interactive controls may distract the operator’s attention from the workplace. Certain types of robots allow defining waypoints via hand guiding. This method uses a special probe on the end effector or data from robot joints to guide the robot by hand to the required position [3]. Another approach is to control the robot from an external application with automatic trajectory generation. This requires additional hardware and software for monitoring the workspace of the robot, motion control and exception handling. Currently, most industrial robot systems use semi-automatic methods for robot programming [2].

In human interaction, verbal and non-verbal communication is intuitively used. Such communication can also be partially used in human-robot communication. This can significantly reduce the time required to train operators of robotic workspaces. In the presented application, we use non-verbal communication, specifically hand gestures, which is a popular way of interacting with or controlling machines, and it has been implemented in many applications, as shown in a comprehensive overview that analyses human-machine interfaces in the smart factory [4]. We focus on the basic operations used to edit robot trajectories for applications that do not require high accuracy. In this article, we compare standard (conventional) ways of defining robot trajectories with our developed gesture-based system in conditions of a real workspace closely resembling the intended industrial application. In the described experiment, the user is able to create, confirm, or delete waypoints, edit their orientation, and move the robot through the defined path.

The current article is organised as follows: In Section 2, an analysis of the State-of-the-Art related to the subject matter is conducted. Section 3 outlines the workspace and principles used in the implementation. Section 4 outlines the experimental setup, hypotheses, and results. In Section 5, the experiments and results are discussed and critically analysed. Finally, Section 6 offers a conclusion of the study, as well as future directions for research.

## 2. Related Work

This section contains an overview of approaches to improve communication in human-robot collaboration by helping robots to understand human intentions as a means of allowing simpler programming and interaction.

### 2.1. Communication Strategies for Intuitive Interaction

People use verbal and non-verbal communication to share information with each other, and these methods can also be employed for human-robot communication. Verbal communication involves the use of voice commands [5,6,7], which can be recognised and acted upon by the robot. This can be achieved through pre-defined commands that execute pre-programmed instructions. Non-verbal communication, on the other hand, involves the use of gestures and facial expressions, with one approach being the tracking of eye movements. By tracking the gaze point and movement [8], the system can determine what object the human is looking at and respond accordingly. Human–robot communication may cover a wider range of applications, with different gesture types triggering specific instructions [4]. In addition to direct robot control, human–robot communication can also encompass safety features. For example, the robot may detect the position of a human and avoid any potential collision [9,10]. Voice commands allow for a wider range of instructions to be programmed compared to gestures, but in noisy industrial environments, voice commands may be distorted and not correctly recognised. Therefore, gestures can be a preferable method of communication for conveying movement information in such environments [2].

### 2.2. Hand Detection and Gesture Recognition Approaches

Hand detection and gesture recognition are important tasks in the field of computer vision and have numerous applications in areas such as human-computer interaction, sign language recognition, and robotics [11,12]. One group of approaches is based on traditional computer vision techniques, such as template matching and feature extraction. These methods rely on handcrafted features and typically require a lot of domain knowledge and manual tuning to achieve good performance. Another group of approaches is based on deep learning techniques, such as convolutional neural networks and recurrent neural networks. These methods have achieved state-of-the-art performance in hand detection and gesture recognition tasks. An example of such a system is MediaPipe Hands [13], which utilizes RGB data to locate the key points of the hand and track their movement. Specialized sensors, such as Kinect [14,15] and Leap Motion [16,17], utilise these approaches and offload the main processing to themselves, allowing for real-time hand detection. These systems also often use depth image data [18] or multimodal data in order to provide more accurate predictions. Training neural network models, however, requires large datasets, which are hard and expensive to acquire [19].

Another group of approaches for hand detection and gesture recognition is based on using gloves [10,20,21,22] and wristbands [23,24] equipped with sensors, tracking devices, or motion-capturing markers such as magnetic or infrared markers. Gloves equipped with sensors or tracking devices can capture data, such as hand position, orientation, and movement, which can be used to identify and recognize gestures as shown in a comprehensive overview that analyses commercial smart gloves [25]. These types of systems are often used in virtual reality and gaming applications, where the user’s hand movements need to be accurately tracked and translated into virtual actions. Motion-capturing markers, such as magnetic [26] or infrared markers [27], are often used in research applications where a high level of accuracy is required. The captured data can then be analyzed to recognize specific gestures or hand poses [28]. Overall, gloves equipped with sensors, tracking devices, or motion-capturing markers can provide highly accurate data for hand detection and gesture recognition tasks. However, they can also be more cumbersome to use compared to other approaches and may require specialized equipment and setup.

### 2.3. Application of Gesture-Based Control

The detection of a hand and recognising of its gestures enables the creation of logic to associate commands with the robot. Existing gesture control applications for robots can be divided into two categories: the control of mobile robots and the control of industrial manipulators. Müezzinoğlu et al. [29] presented a system for real-time control of UAVs using sensor-equipped gloves that detect gestures and hand rotation. They use three different gestures (fist clenched, palm open, and palm clenched with thumb out) in combination with hand tilt information to produce a total of 25 types of commands that represent the direction and speed of movement. A similar application was explored by DelPreto et al. [30], which utilized electromyography-based sensing system for gesture recognition. Carneiro et al. [21] used gestures to teleoperate a walking robot, while Roda-Sanchez et al. [23] navigated the robot to pre-programmed positions using position sensing and wrist rotation. Liu et al. [11] presented a system for modifying a robot’s path using gestures: the application allowed editing the position of points of an already defined path. The system is based on the use of the Leap motion sensor. Four hand gestures are captured (pointing gesture to select a point, swiping a finger on a fictitious line to create a straight line, scissors gesture to cut a curve at a point, drag to catch and move the created point, clenched fist to stop the last operation) [11].

Zhang et al. controlled a robot with a parallel structure using the Leap motion sensor to sense the gesture and hand position. The hand motion (x, y, z) is evaluated, and this motion is transmitted to the parallel structure. The hand motion is complemented by an open-close palm gesture to control the effector closure [12]. Kamath et al. [14] demonstrated real-time control of a mobile robot using upper limb gestures recognised by a Kinect sensor: hands aligned with the body indicate a stop state, while the arms extended forward denote a command to move forward, etc. Quintero et al. [15] presented how Kinect data can be used to calculate the pointing beam coming from a human hand and used to point at objects. The coordinates of the point are calculated as the intersection of the beam and the object. It is worth noting that gesture control has a wide range of applications beyond just human-robot collaboration. For instance, it can provide a solution for privacy-related issues in everyday use of voice assistants [31], be used in healthcare applications [32], and facilitate other forms of human-machine interaction tasks.

The present study introduces a novel system for defining a comprehensive robot path solely through hand gestures without the need for any additional input devices. The proposed approach utilises RGB-D depth cameras to determine the position of the hand while avoiding the use of any extra hardware for tracking hand movements, such as IMU wristbands or gloves. Prior research in this field has shown the utilisation of gestures for either modifying pre-defined paths or directly guiding the robot towards a specific location. For instance, an operator may use gestures to select a point on the path and move it relatively via a drag gesture. Conversely, our system allows for direct entry of points into the path.

## 3. Materials and Methods

This section describes the implementation of the proposed system for defining robot path by gesture commands: the basic concept of our system for controlling the robot by gestures; the workspace on which the system was developed and tested; the system architecture.

### 3.1. Concept

The proposed system’s core component is a monitoring system for a collaborative workspace, which employs multiple RGB-D sensors to ensure comprehensive coverage of the workspace during its operation. The algorithms responsible for processing depth and RGB images from the sensors can identify hands within the space, determine their positions relative to the robot, and recognise their gestures. The hands are represented by the positions of their key points. As the hand can be occluded by the robot body in a single camera view, we opted for a modular system that allows us to add any number of cameras to ensure accurate mapping of hand key points, even under conditions of partial or complete occlusion by the robot. By adding more cameras, we can cover an arbitrarily large workspace. Moreover, each camera provides the added benefit of refining the recognised position of key points. By utilising the recognised gestures and known hand key point positions, waypoints in space are defined, which, in turn, create the robot’s path in a visualisation environment that includes a digital twin of the workplace. This environment also permits direct control of the robot, with commands executed based on gestures performed with both hands.

### 3.2. Workspace

The presented gesture-based path planning system was implemented at our experimental workstation with the Universal Robots UR3e collaborative robot (Figure 1).

This robot has 6 degrees of freedom, an operating range of 500 mm, and a payload of 3 kg. The workstation is made up of a modular system of ITEM aluminium profiles. The work surface comprises an aluminium plate (10 mm thick) with M5 mounting holes. These holes were used to fix the test points in pre-defined positions (more on that in Section 4.1). The robot’s workspace is monitored by three RGB-D Intel RealSense D435 sensors using infrared stereo vision for depth image acquisition. Calibration of camera positions relative to the robot (extrinsics) is performed by detecting the custom grid board with visual markers placed on the workstation [33].

### 3.3. System Architecture

The system was distributed to multiple computational units Jetson Nano due to the high requirements for processing data from each camera. According to the requirements for communication between several devices, it was decided to use the modular architecture offered by the robot operating system (ROS) and to divide the software implementation into several separate components.

The architecture of the system is shown in the diagram in Figure 2. The system is an improved version of the technical solution for mapping obstacles in the robot workspace [10]. The initial data collection from the camera is provided by the Obstacle Mapper block, which packages RGB and depth images together with the intrinsic and extrinsic camera parameters into a ROS message. The message is then published by this block. The modified data are further processed by the Hand Recogniser block, which outputs information that describes the hands captured by the camera. Due to the computational complexity of the hand key points recognition task using MediaPipe Hands [13], the recognition is performed on Jetson Nano minicomputers using their integrated GPUs. Each unit processes data from a single camera. The processed data are then sent to the main Jetson Nano aggregation unit, which performs filtering and merging of the processed data. The output of this unit is the merged information of at most one right-hand and one left-hand. The high-level operation of the system, including GUI, path definition, and its visualisation, is provided by an application running on the operator’s PC. It receives the hands’ data from the main Jetson unit through UDP communication.

**Obstacle Mapper.** It is a ROS node written in C++ using the Intel RealSense SDK. The function is shown in Figure 3. This node reads the raw images (RGB and Depth) provided by the camera. Using the built-in RealSense Align function, the depth image is aligned to the colour image. The node also collects and forwards camera data (intrinsics) and camera position calibration data relative to the robot (extrinsics), including depth scale constant. A detailed description of the calibration of the camera extrinsic in order to express the camera data relative to the robot coordinate system is described in [34]. The ROS message CameraData.msg is constructed from these data and is sent to the Hand recogniser block.

**Hand Recogniser.** This Python node (shown in the diagram in Figure 5) processes the latest message received from CameraData.msg (the queue is set to 1 message) received from ObstacleMapper. The MediaPipe version 0.8.9, which we compiled for CUDA 10.2 [35], is used for hand detection and key points localisation. The use of CUDA allowed us to achieve up to 20 FPS when processing the RGB stream from the camera at 320×240 resolution. The MediaPipe parameters used are shown in Table 1.

In the colour image, the handedness (left or right), hand key points (landmarks), and confidence score are recognised for each hand instance using MediaPipe. The position of the key points is specified as absolute XY pixel coordinates in the 2D image. In the default MediaPipe implementation, the hand key points might be located outside the image boundaries. In order to recognise gestures, all the landmarks must be within the image. Therefore, the points located outside the image boundaries are cropped to the maximum image dimensions. Then, the relative positions of the landmarks to the base point of the palm are calculated; see Figure 4.

Hand gestures are recognised from the relative positions of landmarks using the KeyPointClassifier model provided in [13]. 2D landmarks are projected from the colour image to the depth image using the standard Intel RealSense SDK feature. The obtained 3D coordinates of the landmarks in the camera coordinate system are expressed in the robot coordinate system using extrinsic parameters. The data collected in this node (see Figure 5) are put together into a HandData.msg ROS message that represents information about a single hand. Individual hand messages are combined into a ROS message, MultiHandData.msg, which is sent to the Hand data processor node.

**Hand data processor.** This Python node performs aggregation of data received from individual Jetson Nano units. The function is depicted in Figure 6. The individual MultiHandData.msg messages received from Jetson Nano units are synchronised using the AproximateTimeSynchronizer with a maximum delay of up to 0.1 s (the queue for AproximateTimeSynchronizer as well as for each Subscriber is set to 1 message so that only the latest messages are processed). Two arrays (right and left hands) are created from the received data. The outliners are removed from each array. Whether a hand is an outliner or not is judged by the centroid (calculated from all key points of the corresponding hand) positions of the hands using z-score. The threshold value of the z-score for hand removal is 2 [37]. All the remaining hands on the same side are combined into one so that the individual landmarks are calculated as the median of the corresponding landmarks of all hands in the array. The confidence score is calculated as the average of the confidence score of the hands in the array. The resulting hand gesture for each side is filtered as follows:If the number of unique detected gestures is the same as the number of detected hands, the gesture of the hand with the highest score is considered the resulting gesture.Otherwise, the gesture that is found most frequently is considered the resulting gesture.

The final left-hand and right-hand data are sent using UDP to the operator’s PC with the visualisation application.

**Path planner, visualiser.** Our DirectX11-based visualisation application written in C++ is running on the operator’s PC. We visualise the work cell, the actual position of the robot and hands, the path waypoints created, and the application controls. The application is controlled using pre-defined combinations of hand gestures. The principal scheme of the application is shown in the flow chart in Figure 7.

The Hand gesture logic block is responsible for evaluating the logic of command invocation through gestures. The application has two modes: editing and validation, which can be switched by pointing to the virtual button located in the workspace. The relationship between gestures and commands is illustrated in Figure 8. Gestures are utilised to generate waypoints that define the robot’s path. If a command is given to drive the robot to a specific location, a motion command is generated and sent to the robot controller through an external communication interface (Universal Robots secondary client) using TCP. The robot executes the command thereafter. The secondary client also reads the robot’s current joint variables, and these data are utilised to display the robot’s current location in the application.

## 4. User Study

To assess the efficacy of the human–machine interface (HMI) developed for shared workspaces, a user study was carried out (refer to Figure 9). The study involved a testing system with 20 participants. The purpose of this study was to verify the effectiveness of the developed HMI in shared workspace conditions.

During the experiment, we collected data that included both objective data measurements of each test cycle and subjective data collected through questionnaires. By analysing these data, we were able to compare the different interfaces and assess whether our research hypotheses were valid.

### 4.1. Experiment Description

To assess the usability of a developed interface for defining a robot’s path, an experiment was conducted with 20 university members who volunteered as test participants. The average age of the participants in the experiment was 27 years. The oldest participant was 42 years old, and the youngest was 21 years old. Two participants identified themselves as female and the rest as male. The group was tasked with defining and validating the robot’s path using three different programming interfaces while their performance was evaluated. The experiment measured the quantitative parameters of the time required to program the path and the accuracy of the waypoint definition, which were statistically analysed to determine the status of the research hypotheses. In addition, each volunteer completed a task-specific questionnaire to provide feedback on the usability of each interface and their perceived naturalness during the task execution. The experiment was conducted without causing any harm to the volunteers. Six participants out of the 20 tested subjects had no prior experience in the field of robotics.

The experimental task was based on defining the waypoints of the robot’s path. These points were represented in the workspace by 3D printed targets (see Figure 10). These points are only used to ensure the consistency of the measurements for each trajectory during each trial by providing the user with a visual reference of where the points are located in space.

In each round, the participant’s task was to define a single path from a set of pre-defined paths (5 paths in total) using each of the three interfaces. The order of the paths was defined randomly. Each path consisted of 3 waypoints: the points could be repeated within different paths but were necessarily different within every single path. The following interfaces were tested:**V1—Teach pendant**: the volunteer defines the path using a standard UR robot teach pendant with a PolyScope GUI.**V2—Hand guidance**: the volunteer presses the release button on the teach pendant allowing him to freely drag the robot to a desired position. The positions are saved using PolyScope GUI.**V3—Gesture-based control**: the volunteer uses the proposed gesture-based robot control system to define the waypoints.

A total of 15 tests were conducted with each volunteer (5 paths defined using each of 3 interfaces). The order of the tested interfaces (V1, V2, V3) was selected at random for each volunteer to mitigate the order effect (practise effect [38]) on the measured parameters. Each input point had to be saved in the system’s memory, and then the entire trajectory was traversed to validate the correctness of the input points. During the task, the total measurement time was recorded, including the position of the saved points. The time measurement was finished after the trajectory was completed and validated by the participant.

All experiments were conducted with the required safety precautions, and all test subjects were made aware of potential risks and instructed on safety-critical behaviours.

The participants had the option to participate or not, and they had the freedom to withdraw from the experiment at any point they desired. Prior to the start of the experiment, all participants were obligated to read and sign a consent form as well as the experimental procedures. Once they agreed to participate, any uncertainties regarding the task and the principles of each interface were clarified by the experimenter.

In addition, each volunteer was given the opportunity to test each interface on three different trajectories that were not utilised in the actual experiment prior to commencing the study.

Each round started with a 3-2-1 countdown so that the volunteer knew when to start timing the experiment. Before the start of each round, the volunteer stood in front of the robot in the starting position. This position has been chosen so that the user in the default position is as close as possible to the conditions in actual operation. During each trial, the following outcomes were possible:If the volunteer has entered all the given waypoints correctly and has subsequently validated the entire trajectory, the task is considered successfully completed.If the robot emergency stop was activated for any reason (during the measurement), the measurement is terminated, and the volunteer repeats the entire measurement of that one particular trajectory. Saved values are replaced by newly measured ones.In case the volunteer incorrectly enters any of the waypoints, the values from this measurement are excluded, and the measurement is considered unsuccessful.

In addition to objective parameters, multiple subjective aspects of interacting with the tested interfaces were mapped. The analysis of subjective findings was based on responses to 21 questions for each interface (and additional 3 questions specifically focused on our gesture-based HMI). The questions consisted of three sections.

The first section consisted of six questions (QTLX1-QTLX6, Table 2) used by an independent agency of the National Aeronautics and Space Administration (NASA) to subjectively evaluate newly developed systems. The NASA Task Load Index (NASA-TLX) [39] has been used worldwide to help researchers evaluate workload in various human-machine systems. It is a widely used tool to subjectively evaluate a wide range of areas, from systems to work teams themselves. The results of the test reflect the level of worker workload on a scale of 0 to 100 (where 0 is the lowest load and 100 is the maximum load). The results of the test reflect the level of worker workload on a scale of 0 to 100 (where 0 is the lowest load and 100 Is the maximum load). In our case, a shortened version of this test called the “Raw TLX” was used [40].

The second part consists of 10 questions (QSUS1-QSUS10, Table 3) from the widely used tool System Usability Scale (SUS) [41], which is designed for the subjective evaluation of industrial systems. SUS analysis helps to achieve industry standards and has been used in more than 1300 publications. Subjective questions are rated on a score scale from 1 to 5. From these scores, an overall rating of 0–100 is obtained according to the procedure. SUS provides a single number that represents a composite measure of the overall usability of the system studied.

The last part of the questions consisted of 5 questions related to all interfaces and an additional 3 questions specifically focused on our gesture-based HMI. We evaluate the last section of the questions based on a 1–5 Likert scale (scaling from 1—“totally disagree” to 5—“totally agree”). The questionnaire items were inspired by the works of Bolano et al. [42] and A. Hietanen [43]. However, due to differences in the tested interfaces, the questions have changed significantly.

The first five questions (Q1–Q5, see Table 4) were aimed at comparing aspects of collaboration during task execution by all tested interface variants (V1, V2, V3) and testing the general clarity of the instructions provided. These questions are task and awareness related.

The last three questions (QGB1–QGB3, see Table 5) were task- and awareness-related and aimed to only map work with our HMI (Gesture-based interface). The goal was to find out how the volunteers responded to our HMI and whether the tools used to visualise the digital twin of the system were appropriately chosen.

The volunteer could additionally leave a free comment about any topic related to each interface option.

### 4.2. Hypotheses

The initial hypotheses are based on the suggestion that our proposed robot control interface should make robot trajectory programming more efficient. The dependent measures (objective dependent variables) were defined as task completion time and task accuracy of trajectory point input. The within-subjects independent variable was defined as interface:V1: Teach pendant.V2: Hand guidance.V3: Gesture-based interface.

Overall, it is expected that the test subjects will perform with lower task completion times, similar or lower accuracy of waypoint input, and will have higher subjective ratings when working with V3 Gesture-based interface. The experimental hypotheses were defined as follows:

**Hypothesis 1** **(H1).***The efficiency of the test subjects will be higher with an interface using gestures (V3) than with an interface using a teach pendant (V1) or manual guidance (V2). Higher efficiency is categorised as lower task completion time*.

**Hypothesis 2** **(H2).***The accuracy of creating waypoints by test subjects will be similar or lower for the interface using gestures (V3) as for the interfaces using teach pendant (V1) and manual guidance (V2). The accuracy is categorised as a small deviation of the trajectory waypoint positions*.

**Hypothesis 3** **(H3).***Volunteers will subjectively percept the proposed interface using gestures (V3) as easier to use and more applicable to the task at hand compared to interfaces using a teach pendant (V1) or manual guidance (V2)*.

Hypotheses H1 and H3 are based on our assumption that defining the robot’s path using gestures is more intuitive, more comfortable, and simplifies the task. Hypothesis H2 assumes that hand detection may have an inaccuracy, and this will be reflected in the definition of the waypoints.

### 4.3. Results—Objective Evalutation

Hypothesis H1 states that the efficiency of the test subjects will be higher with interface V3 than with interfaces V1 and V2. The total time taken to complete the tasks for each interface was measured and compared between them. It can be seen from Figure 11 that the average task completion time is significantly higher in V1 condition than V2 and V3 conditions. Completing the task using hand guidance took twice as long as using gestures, a task performed using a teach pendant took four times longer. The completion time means were compared using the ANOVA test, and statistically significant differences were found, F (2.38) = 328.08; *p* < 0.00001.

The following post-hoc *t*-tests revealed statistically significant differences in mean task completion times between V1 (M = 98.05, SE = 3.68) and V2 (M = 51.40, SE = 2.75) interfaces, t (19) = 19.53, *p* < 0.00001. There was also a significant difference in mean task completion times between V1 and V3 (M = 23.69, SE = 0.49) interfaces, t (19) = 20.60, *p* < 0.00001. Thus, hypothesis H1 was supported.

Hypothesis H2 states that the accuracy of entered waypoints for test subjects will be lower with V3 than with V1 and V2. The average deviations of waypoint accuracy were measured. It can be seen from Figure 12 that the average deviation was significantly higher for V3 interface (M = 0.0408, SE = 0.0019) compared to V1 (M = 0.0109, SE = 0.0005) and V2 (M = 0.0110, SE = 0.0007).

The average deviations were compared using ANOVA test. Statistically significant differences were found, F (2,38) = 0.00594; *p* < 0.00001. The following post-hoc *t*-tests revealed a statistically significant difference in mean average deviations between V1 and V3 interfaces, t (19) = −16.16, *p* < 0.00001. Additionally, the difference in mean average deviations between V2 and V3 was statistically significant, t (19) = −15.44, *p* < 0.00001. Thus, hypothesis H2 was supported.

If we express the deviations in the individual axes of the robot coordinate system (see Figure 13), we can observe that the total deviation in the V3 interface consists mainly of the deviation in the Z direction.

### 4.4. Results—Subjective Evaluation

Hypothesis H3 assumes that volunteers’ subjective perception of the proposed gesture-based interface (V3) is easier to use and better suited to the task than the V2 and V1 interfaces. The results of subjective perception consist of three parts.

The first part reports the overall results from the official NASA Task Load Index for each of the interfaces used. It can be seen from Figure 14 that the average task load of the participants in the V3 condition is lower than in the V2 and V1 conditions.

The second part is the overall results from SUS (see Figure 15) for each of the interfaces shows that the average subjective usability of the system in the V3 state is higher than in the V1 state. The data obtained did not indicate sufficient difference for V2 and V3. Therefore, the hypothesis was supported only partially.

The third part of the questions consisted of 5 questions for estimating the subjective comfort and safety of the participant and 3 questions specifically focused on the Hand gesture interface. The first five questions indicate a better subjective feeling of workers when working with our HMI than when working with teach pendant. However, this claim is not sufficiently supported statistically (see Figure 16).

The remaining three questions (QB1–QB3) were not statistically evaluated since their values were obtained only for V3. However, in general, the values indicate the appropriateness of the chosen gestures as well as support the idea of increasing awareness of the future trajectory of the robot (see Figure 17).

## 5. Discussion

In order to assess the developed system, we critically analyse its features and limitations. The presented system uses an innovative type of control interface for industrial robots based on hand gesture detection. During operation of the system, the user does not need to hold or wear any tracking device.

The evaluation of the objective parameters indicated statistically significant evidence that confirms hypotheses H1 and H2. Participants in the experiment were able to define a given path objectively faster using our proposed gesture-based interface. There were four trials during which the participants confused waypoints; since these were significant errors that would have significantly affected the measurements, these trials were considered invalid, and volunteers repeated these trials. The experiment confirmed the predicted higher inaccuracy in defining path waypoints by gestures. The deviation of the inaccuracy of the point input is mainly due to the component in the Z-axis direction in the robot’s coordinate system. We believe that this deviation, with respect to the position of RGB-D sensors, is mainly due to errors in the calculation of RGB depth image projections, specifically noise and low accuracy of depth images. The solution could be the use of a more accurate depth sensor based on the principle of lidar or Time-of-Flight sensors [44].

The following subjective conclusions can be drawn from the questionnaires completed by all participants in the experiment. Interface V3 achieved the best rating among the other tested interfaces in the NASA Task Load Index and in the System Usability Scale, thus partially supporting hypothesis H3. For most of our three custom questions, focussing only on the presented system, no statistical significance of the results of the answers was found. In general, the participants rated the selected gestures for control as appropriate. The visual feedback system chosen in the form of a digital twin of the robot was also positively evaluated. Overall, the graphical interface provided participants with a good understanding of the future path of the robot. Although the results of the subjective evaluation are not definitive, they suggest the potential of using gestures to increase efficiency in defining robot trajectories and, therefore, further application research in this area is worthwhile. The summary of the obtained results is presented in Table 6.

According to the results of the experiment, the creation of waypoints using hand gestures is objectively less accurate than the existing methods for programming robot trajectories. It should also be pointed out that during the experiment participants did not set the orientation of the path waypoints, although the system allowed it. The current results show that the system cannot be used for robot operations that require high precision. Some compromise between speed and accuracy of trajectory input could be achieved by combining our system with existing programming methods. Overall, the experimental results indicate the usefulness of the developed system, as it can speed up the initial definition of the coarse robot path. Data obtained during experiment are available in Appendix A.

## 6. Conclusions

In this paper, we present a robot–human interface that simplifies and streamlines the process of defining a robot path. Using gestures for communication is a natural and intuitive process for humans. The developed system introduces a novel method of control, where the operator can initiate a command for the robot using gestures and directly specify the location for a particular task. The proposed interface was implemented as a part of a robot workspace monitored by three RGB-D sensors which detect the hands of the human worker in the robot workspace. The system relies on distributed data processing using multiple Jetson Nano units, each of which processes data from a single camera. MediaPipe solution is used to localise the landmarks of the hands on the RGB image from which gestures are recognised. The landmarks are then projected onto a depth image to obtain landmarks in 3D space relative to the camera coordinate system. By applying extrinsic parameters and transforming the coordinates, the position relative to the robot’s coordinate system is obtained. The position of the landmarks and hand gestures are used to create, edit, and delete waypoints, and to control the robot traversing along the trajectory. The human worker is provided with visual feedback on the shape of the trajectory through a graphical interface displayed on a monitor located behind the robot.

An experimental study was conducted to compare our developed system of robot programming using gestures with classical approaches using teach pendant and manual robot guidance. The experimental analysis involved the collection of objective quantitative data and subjective qualitative data. It was found that the system of defining the robot trajectory using gestures allows to speed up the process of defining the robot path when programming the robot. The experiment indicated that the current implementation of the system exhibits lower accuracy of waypoints definition compared with the conventional approaches. However, this outcome is defined by the hardware, camera calibration, and potentially the operating conditions, which can be improved by further tuning of the system. We assume that the inaccuracy of the waypoint input is mainly due to the component in the Z-axis direction in the robot’s coordinate system. In this research, we did not focus on high accuracy, as the developed system represents a means for defining robot path “coarsely” for tasks not requiring millimetre accuracy while also providing the user with an intuitive and simple-to-use interface. The gesture-based interface achieved the best rating among the other tested interfaces in the NASA Task Load Index and in the System Usability Scale. In general, the participants rated the selected gestures for control as the most convenient. The visual feedback system chosen in the form of a digital twin of the robot was also positively evaluated. Overall, the graphical interface provided participants with a good understanding of the future trajectory of the robot. Although the results of the subjective evaluation are not definitive, they suggest the potential of using gestures to increase efficiency in defining robot paths.

Future research will focus on improving the developed gesture-based interface. One of the main directions for future research is to work on improving the localisation of the hands to improve the accuracy of the generation of waypoints. Potential improvements could be achieved by implementing more RGB-D sensors or using different sensor types (such as lidar-based sensors). Another direction of development is to focus on extending the functionalities of the existing application. The list of commands could be extended to include functions that are standard in classical robotic tasks. In the future, the visualisation system can also be delegated to an augmented or mixed-reality device. This would further improve the user’s perception of the robot’s planned trajectory. One solution to reduce the inaccuracy of positioning could be to use the gesture only to approximate the area of interest and then attach the target to a salient feature, such as a corner, edge, bolt head, etc.

## Figures and Tables

**Figure 1 sensors-23-04219-f001:**
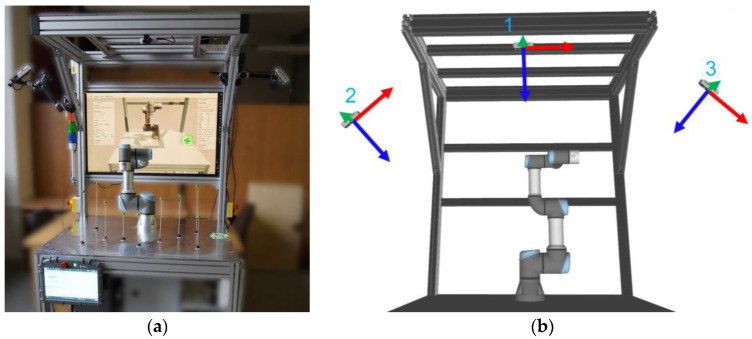
Collaborative workspace with UR3e: (**a**) overview, (**b**) Locations of three RGB-D sensors. The blue vectors depict view directions (z-axes).

**Figure 2 sensors-23-04219-f002:**
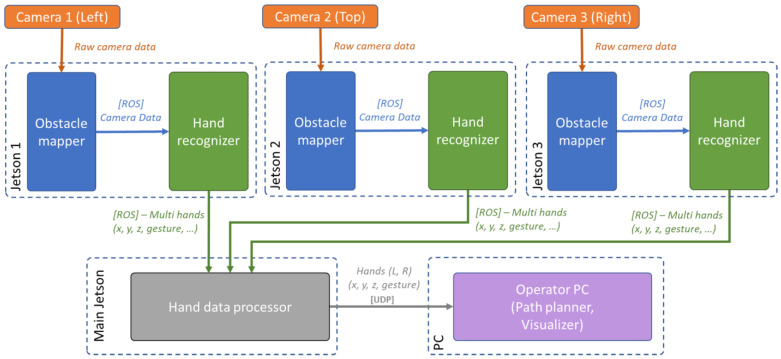
Distributed system providing the gesture-based interface.

**Figure 3 sensors-23-04219-f003:**
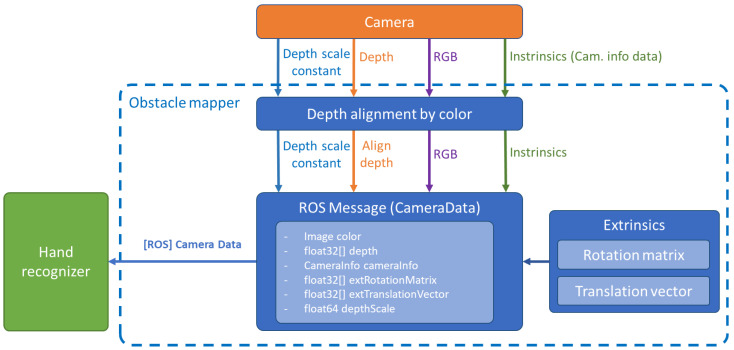
Obstacle mapper data flow chart.

**Figure 4 sensors-23-04219-f004:**
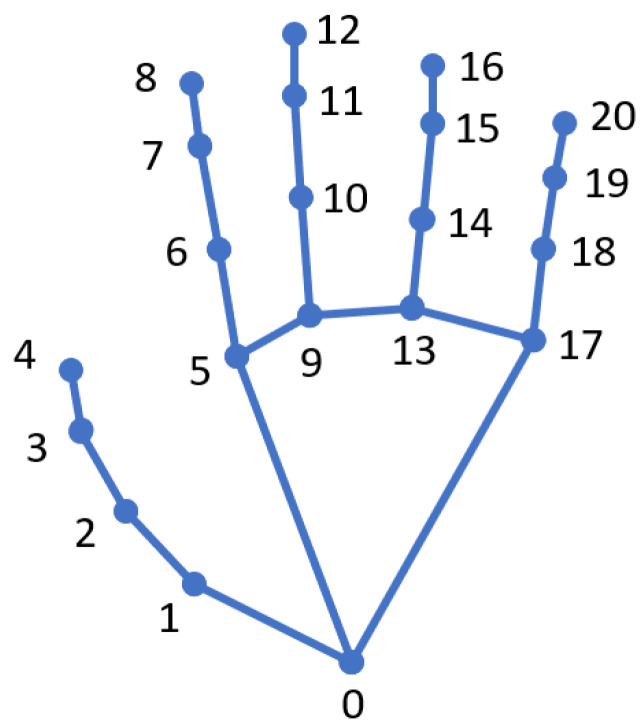
MediaPipe hand landmarks with their indices—hand base point is denoted with “0” [36].

**Figure 5 sensors-23-04219-f005:**
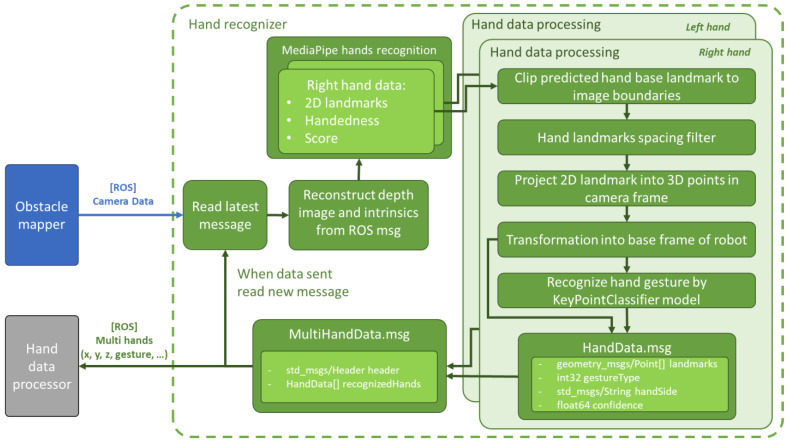
Hand recogniser data flow chart.

**Figure 6 sensors-23-04219-f006:**
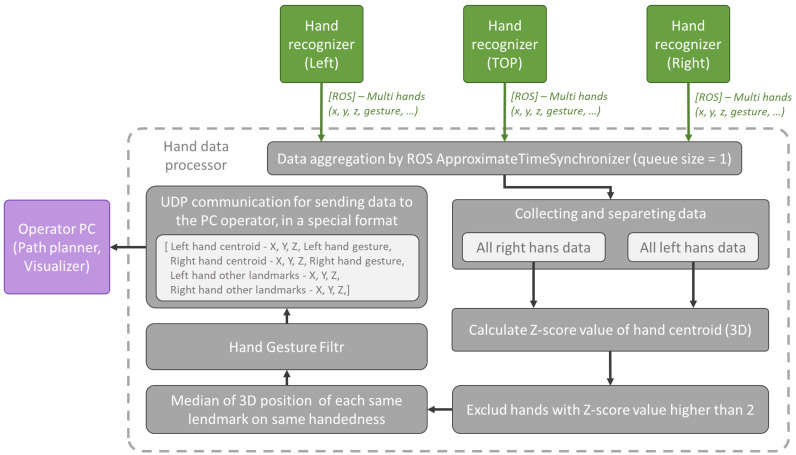
Hand data processor.

**Figure 7 sensors-23-04219-f007:**
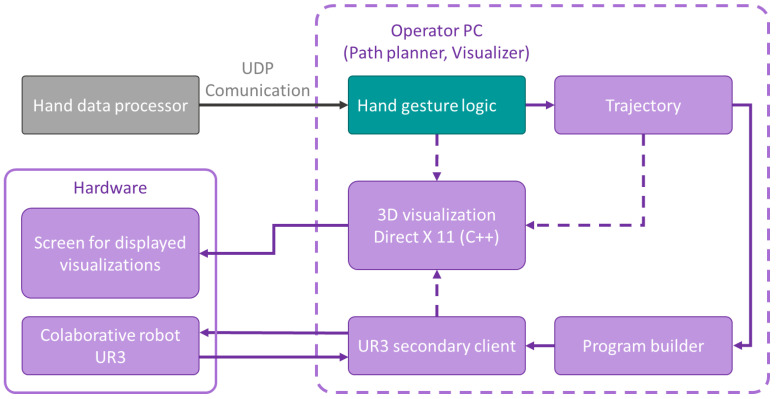
Path planner, visualiser—principal scheme.

**Figure 8 sensors-23-04219-f008:**
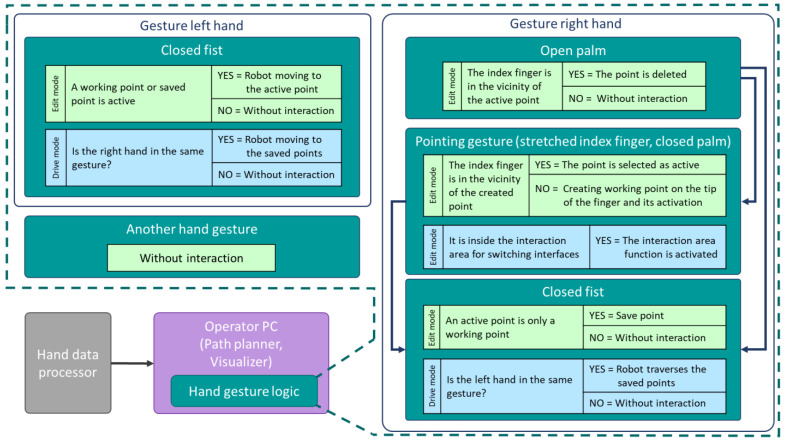
Hand gesture logic—operations activated using left and right hand gestures.

**Figure 9 sensors-23-04219-f009:**
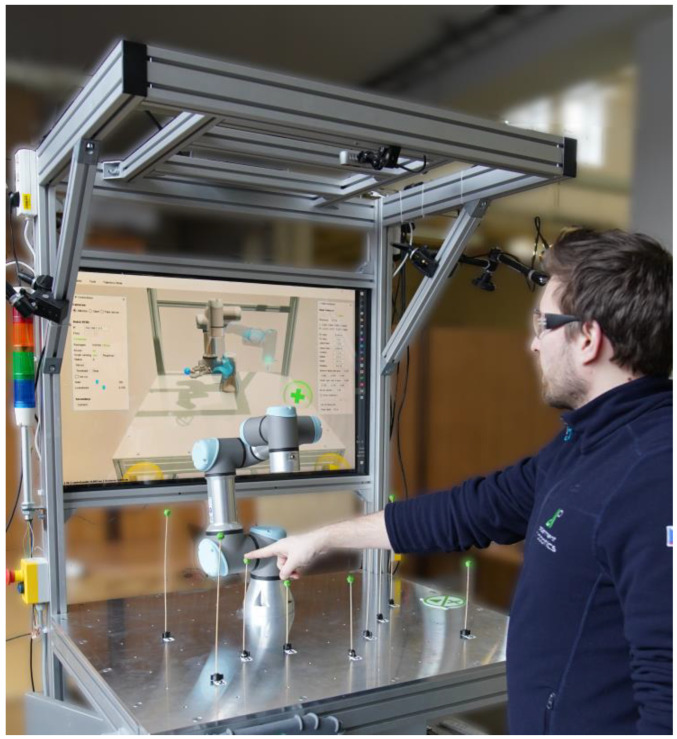
Performed user study: Workspace of the UR3e Robotic Arm with distributed passage points, a screen for visualising the robot’s digital twin and an experiment participant.

**Figure 10 sensors-23-04219-f010:**
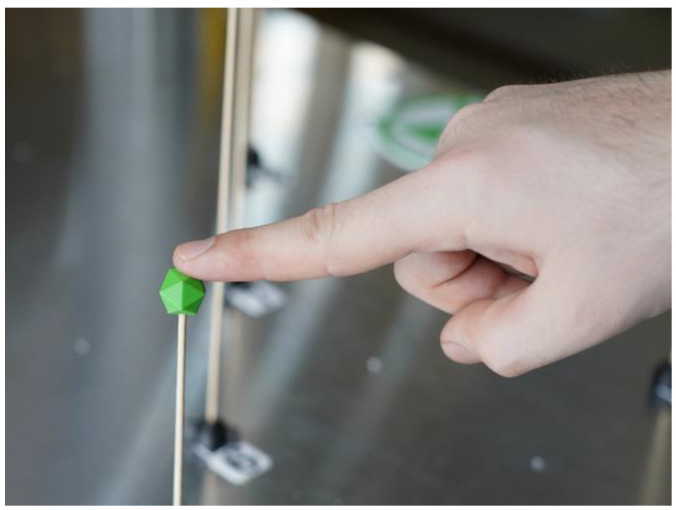
Experiment description: representation of the goal waypoint.

**Figure 11 sensors-23-04219-f011:**
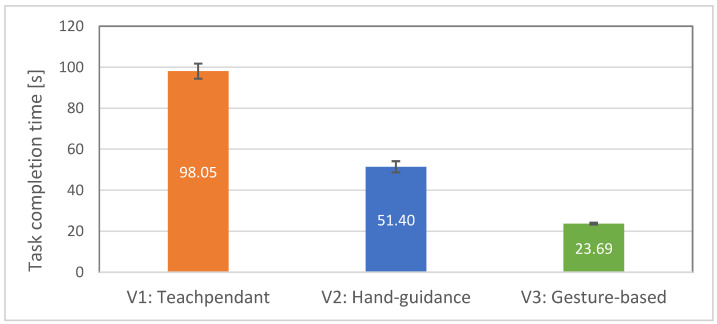
Average task completion time with standard errors for all 20 participants: lower is better.

**Figure 12 sensors-23-04219-f012:**
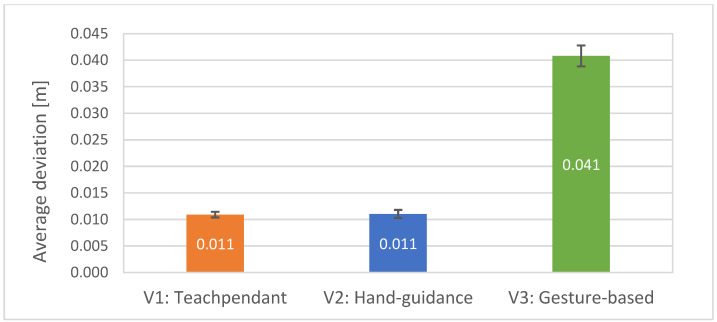
Average deviations of waypoints accuracy with standard errors for all 20 participants: lower is better.

**Figure 13 sensors-23-04219-f013:**
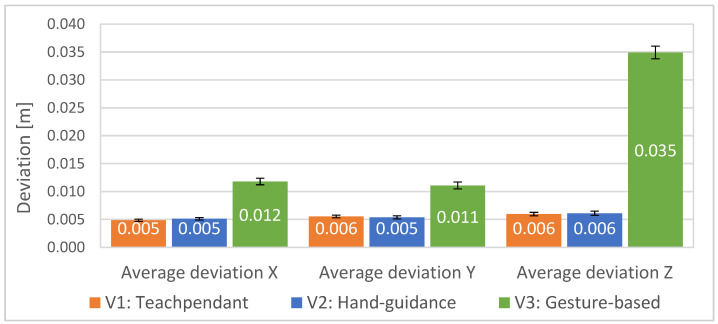
**Figure 13**. Average deviations of waypoints accuracy with standard errors in the individual axis of the robot coordinate system for all 20 participants: lower is better.

**Figure 14 sensors-23-04219-f014:**
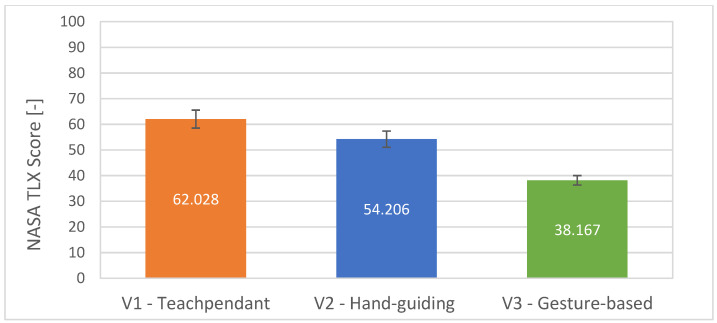
Average scores with standard errors for the questions QTLX1–QTLX6 used for evaluation of the NASA Task Load Index: lower is better.

**Figure 15 sensors-23-04219-f015:**
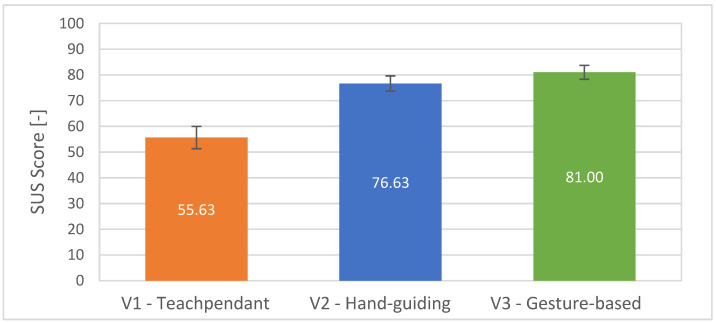
Average scores with standard errors for the questions QSUS1–QSUS10 used for evaluation of the System Usability Scale (SUS): higher is better.

**Figure 16 sensors-23-04219-f016:**
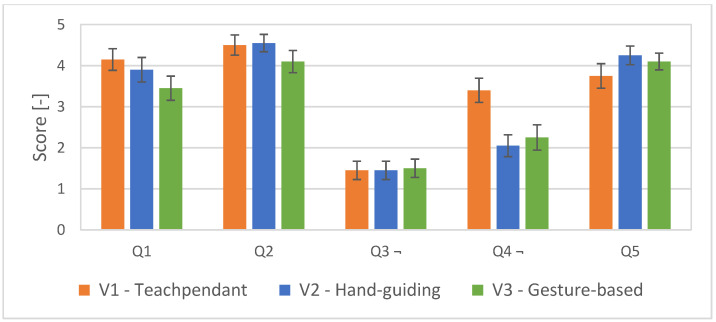
Average scores with standard errors for the questions Q1–Q5. Questions Q1, Q2, Q5—higher is better; Q3, Q4—lower is better (¬).

**Figure 17 sensors-23-04219-f017:**
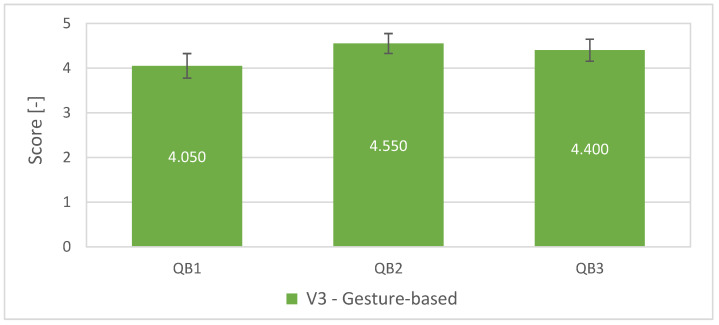
Average scores with standard errors for the questions QB1–QB3.

**Table 1 sensors-23-04219-t001:** MediaPipe Hands parameters.

Parameter	Value
model_complexity	1
max_num_hands	2
min_detection_confidence	0.5
min_tracking_confidence	0.5
static_image_mode	false

**Table 2 sensors-23-04219-t002:** Questions used for NASA Task Load Index for all the tested interfaces (Adapted from [39]).

Task Load Index (NASA-TLX) Questions
QTLX1. How much mental and perceptual activity was required? Was the task easy or demanding, simple or complex?
QTLX2. How much physical activity was required? Was the task easy or demanding, slack or strenuous?
QTLX3. How much time pressure did you feel due to the pace at which the tasks or task elements occurred? Was the pace slow or rapid?
QTLX4. How successful were you in performing the task? How satisfied were you with your performance?
QTLX5. How hard did you have to work (mentally and physically) to accomplish your level of performance?
QTLX6. How irritated, stressed, and annoyed versus content, relaxed, and complacent did you feel during the task?

**Table 3 sensors-23-04219-t003:** Questions used for System Usability Scale for all the tested interfaces (Adapted from [41]).

System Usability Scale (SUS) Questions
QSUS1. I think that I would like to use this system frequently.
QSUS2. I found the system unnecessarily complex.
QSUS3. I thought the system was easy to use.
QSUS4. I think I would need support of technical person to be able to use this system.
QSUS5. I found the various functions in this system were well integrated.
QSUS6. I thought there was too much inconsistency in this system.
QSUS7. I would imagine that most people would learn to use this system very quickly.
QSUS8. I found the system very cumbersome to use.
QSUS9. I felt very confident using the system.
QSUS10. I needed to learn a lot of things before I could get going with this system.

**Table 4 sensors-23-04219-t004:** General questions for all the tested interfaces.

General Questions
Q1. I can do the task better.
Q2. I felt safe operating the robot.
Q3. I felt anxious when working near the robot.
Q4. Controlling the system took my attention away from the workstation. Q5. The system response was comfortable.

**Table 5 sensors-23-04219-t005:** Questions related to Gesture-based interface.

Gesture-Based Interface-Related Questions
QGB1. The HMI was improving my awareness of the robot’s future trajectory.
QGB2. The feedback in the form of displaying a digital twin was easy to understand.
QGB3. The gesture chosen was intuitive for the operation.

**Table 6 sensors-23-04219-t006:** Overview of the experiment results. SUS and questions Q1, Q2, Q5—higher is better; Mean time, mean deviation, NASA TLX, questions Q3, Q4—lower is better (¬).

Parameter/Question	Related Hypothesis	Teach Pendant Results	Hand Guiding Results	Gesture-Based Interface Results	Statistical Significance
Mean time¬ [s]	H1—supported	98.05	51.40	23.69	yes
Mean deviation¬ [m]	H2—supported	0.01	0.01	0.04	yes
SUS (0–100)	H3—supported	55.63	76.63	81.00	yes
NASA TLX¬ (0–100)	H3—supported	62.03	54.21	38.17	yes
Q1 (1–5)	H3—supported	3.45	3.90	4.15	yes
Q2 (1–5)	H3—not supported	4.50	4.55	4.10	no
Q3¬ (1–5)	H3—not supported	1.45	1.45	1.50	no
Q4¬ (1–5)	H3—supported	3.40	2.05	2.25	yes
Q5 (1–5)	H3—not supported	3.75	4.25	4.10	no

## Data Availability

The data presented in this study are available in the Appendix A. Code is available in a publicly accessible repository that does not issue DOIs: https://github.com/anion0278/mediapipe-jetson (accessed on 22 April 2023), https://github.com/anion0278/dms_perception (accessed on 22 April 2023).

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
