# Peer review of "Hand Gesture Interface for Robot Path Definition in Collaborative Applications: Implementation and Comparative Study"

_sensors, 2023, doi:10.3390/s23094219_

Round 1
Reviewer 1 Report
The paper presents a study on a system aimed at using hand gestures as a control interface for robotic systems in a collaborative workspace.
The work is intersting and original enough. The architecture of the system is properly described and the experiments have significance.
However, in order to make the work suitable for publication, the Authors should improve the paper by taking into account the following suggestions:
- the title is not fully consistent with the content of the paper, hence it should be changed into something like "Study of a system for defining trajectories for a collaborative robot by means of hand gestures" (or something similar);
- the subtitles of Section 2 are not consistent with the content of the subsections and have to be changed;
- the title of Section 3 is not consistent, and it should be changed into "Architecture of the system";
- the set of volunteers used in the user's study must be better specified (e.g. age, gender of the participants, etc.). Moreover, a more scientific discussion of the statistic validity of the test should be carried out;
- the presentation of the results should be improved. Several aspects and tests are taken into consideration, and the presentation is not completely clear. I suggest to create a table that gathers all the results, so as to improve the clarity;
- please revise the English language, as there are some syntactical errors.
Author Response
Authors: The authors appreciate all the reviewers and editors for the careful reading of our manuscript and insightful suggestions. After considering the comments carefully, we have revised the manuscript accordingly. The changes applied in reaction to Reviewer 1 are highlighted by the yellow colour in the manuscript revision. Summary of the changes:
Reviewer 1: The paper presents a study on a system aimed at using hand gestures as a control interface for robotic systems in a collaborative workspace. The work is intersting and original enough. The architecture of the system is properly described and the experiments have significance. However, in order to make the work suitable for publication, the Authors should improve the paper by taking into account the following suggestions:
- Reviewer 1: the title is not fully consistent with the content of the paper, hence it should be changed into something like "Study of a system for defining trajectories for a collaborative robot by means of hand gestures" (or something similar);
Authors: Thank you for your comment. We would like to point out that the article describes primarily the approach and implemented system, rather than the experiment. This is the reason why we have chosen the current title. In addition, we have already confirmed this title to the editorial team and we are not sure if a change is possible at this stage.
- Reviewer 1: the subtitles of Section 2 are not consistent with the content of the subsections and have to be changed;
Authors: We have addressed this issue by improving the subsection names.
- Reviewer 1: the title of Section 3 is not consistent, and it should be changed into "Architecture of the system";
Authors: We appreciate your comment. “Materials and methods” is the standard and recommended title of this chapter for MDPI Sensors. We believe that "Materials and Methods" accurately reflects the content of this section, as it covers not only the system architecture, but also the broader concept and resources involved. We used the title “System architecture” for subsection 3.3, which primarily focuses on describing the architecture of the system itself.
- Reviewer 1: the set of volunteers used in the user's study must be better specified (e.g. age, gender of the participants, etc.). Moreover, a more scientific discussion of the statistic validity of the test should be carried out;
Authors: We added basic statistical data about the participants of the experiment (average age and gender) in section 4.1. To evaluate the data obtained during experiment, we used the statistical methods t-test and ANOVA (sections 4.2, 4.3), which allowed us to assess the significance and relevance of the results. Our main goal in this paper was to present the concept and implementation of the system and in our opinion this sample of volunteers and its evaluation is sufficient. We did not observe a statistically significant impact of volunteers' robotics experience, so we did not include this data. We also added complete data from the experiment used in the analysis as the supplementary materials.
- Reviewer 1: the presentation of the results should be improved. Several aspects and tests are taken into consideration, and the presentation is not completely clear. I suggest to create a table that gathers all the results, so as to improve the clarity;
Authors: Thank you for your valuable feedback regarding the presentation of the results. We have carefully considered your suggestion and in response to your recommendation, we have incorporated an additional table in the Discussion section, which summarizes all the relevant results of our experiment. Furthermore, we have included the detailed experiment data in the supplementary material of the paper, ensuring that interested readers can access the complete information.
- Reviewer 1: please revise the English language, as there are some syntactical errors.
Authors: Thank you for your feedback. We have performed additional proofreading in the manuscript revision.
Reviewer 2 Report
The authors explore the possibilities of using hand gestures as a control interface for robotic systems in collaborative workspace. The work is very interesting, with some minor errors listed in sequence. The comments in the next paragraphs are intended to improve paper quality and readers' understanding.
why did you choose to use 3 rgb-d cameras? from the images shown in the paper, it seems that a single top rgb-d camera would be enough.
Please provide more detail regarding how the 3 camera coordinate systems are calibrated so that the data captured by the 3 depth cameras are fused in a single coordinate system. This is not clear in the text.
"conducted with 20 university members who volunteered as test participants." and "The experiment was conducted without causing any harm to the volunteers, and included six participants who had no prior experience in the field of robotics." -> how many users participated on the experiments? 20 or 6?
Please provide more detail regarding how the tests were performed. From Figure 9, it is possible to see different passage points and the user pointing to one of them. Did each user have to randomly select 3 passage points using the teach pendant, hand guidance and gesture? How did the system identify when the user is pointing at an specific passage point? Do you perform any color segmentation based on the green color?
It would be nice to see some videos or more images about the tests performed.
More general comments and minor errors are listed as follows.
"differentiate our system different from" -> "differentiate our system from"
"while providing high . " -> ?
"Kamath et al. demonstrated real time control of a mobile robot using upper limb gestures recognized by a Kinect sensor: hands aligned with the body indicate a stop state, when the arms are extended forward denoted a command to move forward, etc" -> reference is missing
"data flow chart " -> "data flow chart."
"with “0” [31] " -> "with “0” [31]."
"data flow chart " -> "data flow chart."
"data processor" -> "data processor."
"principal scheme" -> "principal scheme."
"with left and right hand " -> "with left and right hand."
please fix word spacing in line 274
"Total of 15" -> "A total of 15"
"The tool is a widely used tool" -> "It is a widely used tool"
"The last three question " -> "The last three questions"
" suggestion is that" -> " suggestion that"
"interfaces used Figure" -> "interfaces used. Figure"
"QB1-QB3" -> "QB1-QB3."
"participates" -> "participants"
" the position of the positioned" -> please rewrite
"assume, that" -> "assume that"
Author Response
Authors: The authors appreciate all the reviewers and editors for the careful reading of our manuscript and insightful suggestions. After considering the comments carefully, we have revised the manuscript accordingly. The changes applied in reaction to Reviewer 2 are highlighted by the green colour in the manuscript revision. Summary of the changes:
Reviewer 2: The authors explore the possibilities of using hand gestures as a control interface for robotic systems in collaborative workspace. The work is very interesting, with some minor errors listed in sequence. The comments in the next paragraphs are intended to improve paper quality and readers' understanding.
- Reviewer 2: why did you choose to use 3 rgb-d cameras? from the images shown in the paper, it seems that a single top rgb-d camera would be enough.
Authors: As the hand can be occluded by the robot body in a single camera view, we opted for a modular system that allows us to add any number of cameras to ensure accurate mapping of hand key points, even under conditions of partial or complete occlusion by the robot. By adding more cameras, we can cover an arbitrary large workspace; moreover, each camera provides the added benefit of refining the recognized position of key points.
To clarify this decision, we have included an explanation in Chapter 3.1 Concept.
- Reviewer 2: Please provide more detail regarding how the 3 camera coordinate systems are calibrated so that the data captured by the 3 depth cameras are fused in a single coordinate system. This is not clear in the text.
Authors: To align coordinate systems, we use the grid board-based aligning method described in the following works:
- Oščádal, P.; Heczko, D.; Vysocký, A.; Mlotek, J.; Novák, P.; Virgala, I.; Sukop, M.; Bobovský, Z. Improved Pose Estimation of Aruco Tags Using a Novel 3D Placement Strategy. Sensors 2020, 20, 4825. https://doi.org/10.3390/s20174825.
- Oščádal, P.; Spurný, T.; Kot, T.; Grushko, S.; Suder, J.; Heczko, D.; Novák, P.; Bobovský, Z. Distributed Camera Subsystem for Obstacle Detection. Sensors 2022, 22, 4588. https://doi.org/10.3390/s22124588
In order to clarify this point in the manuscript, we have added an explanation in Chapter 3.3.
- Reviewer 2: "conducted with 20 university members who volunteered as test participants." and "The experiment was conducted without causing any harm to the volunteers, and included six participants who had no prior experience in the field of robotics." -> how many users participated on the experiments? 20 or 6?
Authors: Thank you for your comment. To avoid possible confusion, we have modified the definition of the test subjects in L313. In the article, we specified that the experiment involved 20 volunteers and that 6 of them had no previous robotics experience.
- Reviewer 2: Please provide more detail regarding how the tests were performed. From Figure 9, it is possible to see different passage points and the user pointing to one of them. Did each user have to randomly select 3 passage points using the teach pendant, hand guidance and gesture? How did the system identify when the user is pointing at an specific passage point? Do you perform any color segmentation based on the green color?
Authors: Each user was tasked with defining a randomly selected trajectory from a set of predefined trajectories in each measurement round. The system does not use any algorithm to recognize the green reference markers from the camera, as these points are only used to ensure the consistency of the measurements for each trajectory during each trial by providing user with visual reference of where the points are in space. Their position was defined relative to the robot base (we also verified the correspondence between expected and real markers by locating them with the robot TCP). During the experiment we evaluate the accuracy of the points defined by a test subject by comparing the created points with the reference position of the corresponding waypoint. We also added this information to the text to avoid confusion.
- Reviewer 2: It would be nice to see some videos or more images about the tests performed.
Authors: We have provided additional supplementary material with video demonstration of the system.
- Reviewer 2: More general comments and minor errors are listed as follows:
- "differentiate our system different from" -> "differentiate our system from"
- "while providing high . " -> ?
- "Kamath et al. demonstrated real time control of a mobile robot using upper limb gestures recognized by a Kinect sensor: hands aligned with the body indicate a stop state, when the arms are extended forward denoted a command to move forward, etc" -> reference is missing
- "data flow chart " -> "data flow chart."
- "with “0” [31] " -> "with “0” [31]."
- "data flow chart " -> "data flow chart."
- "data processor" -> "data processor."
- "principal scheme" -> "principal scheme."
- "with left and right hand " -> "with left and right hand."
- please fix word spacing in line 274
- "Total of 15" -> "A total of 15"
- "The tool is a widely used tool" -> "It is a widely used tool"
- "The last three question " -> "The last three questions"
- " suggestion is that" -> " suggestion that"
- "interfaces used Figure" -> "interfaces used. Figure"
- "QB1-QB3" -> "QB1-QB3."
- "participates" -> "participants"
- " the position of the positioned" -> please rewrite
- "assume, that" -> "assume that"
Authors: Thank you for pointing out these mishaps, we have corrected them and performed additional proofreading in the manuscript revision.
Reviewer 3 Report
This study explores the possibilities of using hand gestures as a control interface for robotic systems in collaborative workspace. The topic of this paper is closely related to the hot topics in the field of the trajectory definition of industrial robots, and content is substantial. But there are a few things that need to be improved.
1. In the abstract and conclusion, the paper does not highlight the new discovery, significance and contribution, and the difference and innovation from other similar work;
2. The paper has a few format problems, such as the format of the reference is different from the standard required by the journal; The position of the picture description is not uniform; The font size of some words in some pictures is too small.
3. If relevant regulations are not violated, the gender, age, education, physical quality and other information of the 20 volunteers should be explained, because these factors may affect the test results.
Author Response
Authors: The authors appreciate all the reviewers and editors for the careful reading of our manuscript and insightful suggestions. After considering the comments carefully, we have revised the manuscript accordingly. The changes applied in reaction to Reviewer 3 are highlighted by the blue colour in the manuscript revision. Summary of the changes:
Reviewer 3: This study explores the possibilities of using hand gestures as a control interface for robotic systems in collaborative workspace. The topic of this paper is closely related to the hot topics in the field of the trajectory definition of industrial robots, and content is substantial. But there are a few things that need to be improved.
- Reviewer 3: In the abstract and conclusion, the paper does not highlight the new discovery, significance and contribution, and the difference and innovation from other similar work;
Authors: We have outlined the unique contributions and distinctions of our work from the previous studies in L144 Furthermore, we have expanded the abstract and conclusion section to integrate this information.
- Reviewer 3: The paper has a few format problems, such as the format of the reference is different from the standard required by the journal; The position of the picture description is not uniform; The font size of some words in some pictures is too small.
Authors: Thank you for your feedback. We have increased the size of the figures to ensure their readability and refined the references format. It is also worth noting that the images in the file are of high resolution, allowing them to be zoomed in on.
- Reviewer 3: If relevant regulations are not violated, the gender, age, education, physical quality and other information of the 20 volunteers should be explained, because these factors may affect the test results.
Authors: In order to provide a more comprehensive description of the experiment, we have expanded the information about the participants involved in the study (L304) and provided the experiment data (experiment_results.xlsx) in the supplementary material. While only general information is presented in this article, more specific data can be found in the attached Excel file.
Reviewer 4 Report
In my opinion, in this work there is not a scientific contribution to the modern state of the art, nor a feasible solution.
The experiment confirmed higher inaccuracy in defining trajectory waypoints by gestures, the creation of waypoints using hand gestures is less accurate than the existing methods for programming robot trajectories, and the system cannot be used for robot operations that require high precision. Research should be directed to use suitable sensors to improve the accuracy of the generation of waypoints.
Author Response
Reviewer 4: In my opinion, in this work there is not a scientific contribution to the modern state of the art, nor a feasible solution.
The experiment confirmed higher inaccuracy in defining trajectory waypoints by gestures, the creation of waypoints using hand gestures is less accurate than the existing methods for programming robot trajectories, and the system cannot be used for robot operations that require high precision. Research should be directed to use suitable sensors to improve the accuracy of the generation of waypoints.
Authors: We appreciate feedback on our manuscript. We would like to thank you for taking the time to provide an opinion. However, we would like to point out that our system was not intended to be part of high-precision applications. Our goal was not to create a system that was highly accurate, as we present our system as a means to quickly define a robot path "coarsely" for tasks not requiring millimeter-accuracy (not all automation applications require such accuracy), while also providing user with intuitive and simple-to-use interface. We have made changes to the conclusion to better highlight the expected field of use of this system. We hope that this will help readers understand the potential applications of our work.
Our experiment points to improvements in speed of defining robot path, increased intuitiveness, and clarity, which are all significant benefits for human-robot collaboration in the manufacturing industry. Our gesture-based interface achieved the highest rating among the other tested interfaces in the NASA Task Load Index and also in the System Usability Scale.
Our future research will focus on improving the developed gesture-based interface. We agree with suggestion that one of the main directions for future research is improving the localization of the hands and its overall accuracy.
Thank you for your valuable feedback, and we hope that you find our manuscript informative.
Round 2
Reviewer 1 Report
The Authors have revised the paper taking into account most of my suggestions. I still believe that the title of the paper, as well as the title of Section 3, should be changed in order to make them more consistent with the contents. Thus, I invite the Authors to proceed accordingly.
Author Response
Authors: After considering the comments carefully, we have revised the manuscript accordingly. The changes applied in reaction to Reviewer 1 are highlighted by the yellow colour in the manuscript revision. Summary of the changes:
Reviewer: The Authors have revised the paper taking into account most of my suggestions. I still believe that the title of the paper, as well as the title of Section 3, should be changed in order to make them more consistent with the contents. Thus, I invite the Authors to proceed accordingly.
Authors: We have taken into consideration the feedback regarding the title of our paper and have made it more descriptive and included the word "study" to also emphasize the experiment. The revised title is: "Hand Gesture Interface for Robot Path Definition in Collaborative Applications: Implementation and Comparative Study". As for section 3, we believe that the current title accurately represents its content, as we explained in our previous response, and have therefore decided to keep it unchanged.
Reviewer 4 Report
The manuscript has been improved and it was clarified that the proposed system was not intended to be part of high-precision applications. Some issues should be addressed before this manuscript could be considered for publication.
Some recent paper (https://doi.org/10.3390/automation2040016), proposed a single camera approach and is verified only by simulation, it is advisable to highlight the advantages of your work.
The experimental study is important to determine the accuracy of the proposed method, and Figure 12 provides a guide for lower precisions applications. I would recommend reviewing Line 559, “we decided to complement the article with an experimental study that compared our developed system of robot programming using gestures with classical approaches using teach pendant and manual robot guidance”.
Perhaps it is unnecessary to state, Line 566 “The experiment also confirmed the predicted higher inaccuracy in defining path 566 waypoints by gestures”.
Author Response
Authors: After considering the comments carefully, we have revised the manuscript accordingly. The changes applied in reaction to Reviewer 4 are highlighted by the green colour in the manuscript revision. Summary of the changes:
Reviewer: The manuscript has been improved and it was clarified that the proposed system was not intended to be part of high-precision applications. Some issues should be addressed before this manuscript could be considered for publication.
Reviewer: 1) Some recent paper (https://doi.org/10.3390/automation2040016), proposed a single camera approach and is verified only by simulation, it is advisable to highlight the advantages of your work.
Authors: Thank you for your feedback, we have carefully considered your suggestion and incorporated changes in our abstract and introduction to emphasize that our system underwent not only simulation-based verification but also experimental verification through its deployment and operation in a real-world setting.
Reviewer: 2) The experimental study is important to determine the accuracy of the proposed method, and Figure 12 provides a guide for lower precisions applications. I would recommend reviewing Line 559, “we decided to complement the article with an experimental study that compared our developed system of robot programming using gestures with classical approaches using teach pendant and manual robot guidance”.
Perhaps it is unnecessary to state, Line 566 “The experiment also confirmed the predicted higher inaccuracy in defining path 566 waypoints by gestures”.
Authors: Thank you for your comment. We have changed the sentences in L566 and L571.